# Tilted Losses in Training Quantum Neural Networks

## Abstract

Empirical risk minimization is a fundamental paradigm in the optimization process of machine learning (ML) models. Several techniques extend this idea by introducing parameters which further regularize this strategy in training these models. One of these paradigms is the so-called tilted empirical risk minimization (TERM), which uses a tilted hyperparameter to penalize the presence of outliers, which represent data samples that differ significantly from the rest of the dataset. Quantum machine learning (QML) models have been studied and benchmarked across various criteria stemming from classical ML, including their training via the parameter-shift rule. Therefore, it is natural to extend the concept of TERM in training QML models, namely the type of models known as quantum neural networks (QNNs). In this work, we examine the impact of a tilted loss function in training a class of QNNs, specifically for binary classification tasks involving two different datasets with induced class imbalance. In the first dataset, the Iris dataset, we show that varying the value of the tilted hyperparameter modifies the decision boundary leading to reduced importance of outliers and better training accuracy — highlighting the importance of using tilted risk minimization. Additionally, in a synthetic dataset we validate that the training accuracy can be improved using the tilted parameter. Analytically, we extend the parameter-shift training method to accommodate weighted inputs by introducing the tilted hyperparameter for training QNNs. These results highlight the significance of incorporating regularization techniques from ML models into QML models.

## 1 Introduction

In the past years, quantum machine learning (QML) has received heightened interest as an application of quantum algorithms and quantum computing more generally (Schuld & Petruccione, 2021; Biamonte et al., 2016; Beer et al., 2020). An important class of quantum learning models are known as quantum neural networks (QNNs) (Cerezo et al., 2021; Farhi & Neven, 2018; Benedetti et al., 2019; Havlíček et al., 2019; Schuld et al., 2020; McClean et al., 2016). QNNs are also known to be variational quantum algorithms (VQAs), or parameterized quantum circuits (PQCs) due to their parameterized nature, which allows them to be trained on classical data using optimization techniques to minimize a loss function. While QNNs are expected to have potential, they face notable challenges, particularly in their training processes, as they are prone to the barren plateaus problem, where the training landscape resembles a flat structure with gradient values leading to zero (Larocca et al., 2024; McClean et al., 2018).

Characteristics such QNN trainability, generalization, expressivity and interpretability of these models have been extensively studied (Mitarai et al., 2018; Du et al., 2020; Abbas et al., 2021; Schuld et al., 2021; Banchi et al., 2021; Pira & Ferrie, 2024). In particular, training methods are brought forth and studied in response to the success of backpropagation for classical architectures (Mitarai et al., 2018; Schuld et al., 2020; Beer et al., 2020; Abbas et al., 2023; Rumelhart et al., 1986). Parameter-shift rule is one such training method that incorporates ideas similar to the chain rule adopted by backpropagation (Mitarai et al., 2018; Schuld et al., 2020). Albeit, plagued by caveats, including its scalability (Hubregtsen et al., 2022; Kottmann et al., 2021)

Quantum learning models are often directly inspired by the architectures of classical machine learning (ML) models (Goodfellow et al., 2016; LeCun et al., 2015; Bishop, 2006). The idea is based on

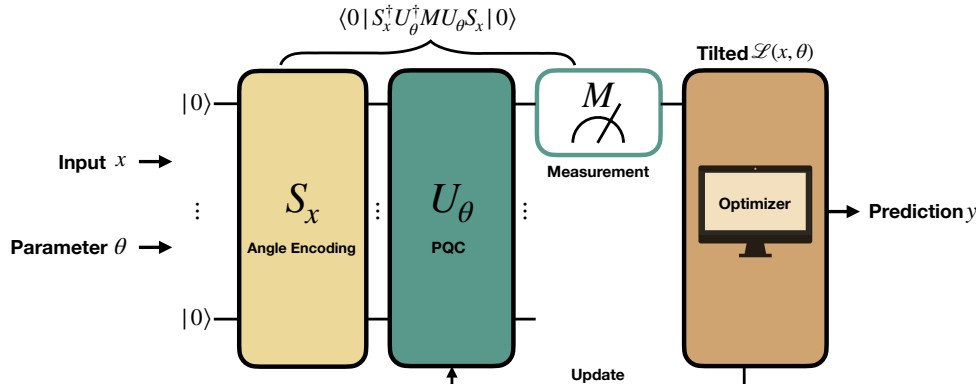

Figure 1: *Illustration of the QNN structure with the tilted loss training.* For training, input $x$ is encoded in $S_x$ via angle encoding for state preparation. The computation proceeds through the parameterized quantum circuit (PQC) with $U_\theta$ as unitary transformation on parameter $\theta$. The expectation value $\langle 0|S_x^\dagger U_\theta^\dagger M U_\theta S_x|0\rangle$ is obtained after measurement and passed to classical optimizer with tilted losses. Finally, the model outputs the prediction $y$.

the widespread success of ML models (Krizhevsky et al., 2012; He et al., 2015), making it a convenient starting point for exploring learning models through the quantum lens. However, classical models also encounter training and generalization difficulties, especially in addressing biases and handling outliers. To address these challenges, a segment of the literature proposes techniques to mitigate bias and handle outlier data (Goodfellow et al., 2016).

The premise of optimization in ML models is carried out under the framework of empirical risk minimization (ERM), whereby the idea is to minimize the average loss across the training dataset (Vapnik, 1998; Block et al., 2024). However, ERM has many practical shortcomings, such as overfitting to outliers and producing biased solutions that may unfairly impact certain subgroups. To address these shortcomings, there exist other alternative methods that modify ERM. One such example is tilted empirical risk minimization (TERM) (Li et al., 2020; 2023), which explores the concept of tilted losses in machine learning for risk minimization. Despite exponential tilting being a well-known tool in statistics (Siegmund, 1976; Butler, 2007), information theory (Thomas & Joy, 2006) and applied probability (Dembo, 2009), it has only recently gained traction in machine learning, as TERM offers flexibility in adjusting loss functions to improve fairness and robustness. For logistic regression, TERM offers a more flexible approach. Ref. (Li et al., 2023) proposes TERM modifies the slope of the decision boundary instead of adjusting the threshold, allowing for better handling of misclassified data and more challenging datasets. This flexibility enables TERM to outperform traditional classifiers like logistic regression, particularly in scenarios involving noisy data and class imbalance. For completeness in this line of literature, structural risk minimization (SRM), first set out in Ref. (Vapnik & Chervonenkis, 1974), introduces a penalty term for model complexity to prevent overfitting and improve generalization to unseen data. Additionally, quantile risk minimization (QRM) optimizes over specific quantiles to reduce the influence of extreme outliers (Eastwood et al., 2022). These techniques aim to produce more robust models by minimizing risks associated with overfitting and bias, while improving generalization.

**Our contributions.** We analyze how regularization strategies from classical literature apply to the trainability of quantum models, specifically QNNs, given the classical-quantum correspondence noted above. Our work investigates the impact of tilted loss on the training of this class of QNNs. To our knowledge, TERM has not been previously explored in the context of variational quantum architectures. Our work is the first to analyze the impact of TERM for training QNNs, the method shown in Fig 1, demonstrating its potential to enhance model performance, particularly under class imbalance and outliers. Our contributions can be summarized as follows.

- We analyze the effects of standard ERM and TERM on the decision boundary while training QNNs using the Iris and Synthetic datasets (see Fig 3). We observe that TERM significantly enhances the training process for classification tasks. Specifically, training with

tilted loss leads to better decision boundaries compare to ERM, particularly in the presence of class imbalance and outliers. Experimental results on both datasets demonstrate the effectiveness and flexibility of TERM for handling classification with data imbalance.

- Moreover, with fine-tuned tilted hyperparameters, TERM outperforms ERM in QNN classification tasks on both datasets, improving accuracy by ~8% (see Table 1).

- These observations motivate us to propose a new algorithm to train QNNs based on extending the parameter-shift rule to tilted losses. This new algorithm optimizes the training process by leveraging tilted loss to learn potentially weighted training data. This approach is important for robust learning in scenarios where datasets are corrupted by noise or other imperfections.

**Related works.** Similar regularization techniques have started to be explored in the QML context. For instance, SRM is applied analytically to two quantum linear classifiers in Ref. (Gyurik et al., 2023), highlighting the trade-off between training accuracy and generalization performance in parameterized quantum circuits. Ideas on quantum learning for quantum data have been brought forth in Ref. (Heidari et al., 2021), hereby formalizing ideas on quantum ERM. Additionally, Ref. (Heidari & Szpankowski, 2024) introduces a quantum ERM algorithm that improves sample complexity bounds in quantum settings through quantum shadows, enabling more efficient empirical loss estimation in quantum classifiers. Moreover, Ref. (Ciliberto et al., 2020) addresses the question of bounds from a quantum learning perspective. Arunachalam and de Wolf's survey in Ref. (Arunachalam & De Wolf, 2017) inspects quantum adaptations of classical models like PAC (probably approximately correct) learning, noting challenges such as sample complexity and measurement incompatibilities due to the fundamental nature of quantum mechanics.

**Outline.** This manuscript is structured as follows. Section 2 presents preliminary information on the TERM paradigm, and training of QNNs. In Section 3 we present the numerical experiments based on the two datasets. Section 4 formalizes parameter-shift rule with a tilted hyperparameter for calculating quantum gradients. This study concludes in Section 5 with a summary of its contributions and open problems.

## 2 BACKGROUND

In this section, we introduce TERM and QNNs. QNNs, based on parameterized quantum circuits, perform binary classification by optimizing quantum parameters.

### 2.1 TILTED EMPIRICAL RISK MINIMIZATION

Empirical risk minimization is widely used in machine learning where the goal is to optimize model parameters by minimizing the average loss over the training data (Vapnik, 1998). The key idea behind ERM is that, because we do not know the true distribution that generated the data, we use the available training data to estimate the risk by averaging the loss over the dataset. However, ERM has notable limitations, particularly when dealing with outliers or imbalanced data, and generalizing to unseen data. For example, in situations where certain data subgroups are underrepresented or contain outliers, the model may overfit to noisy data or produce unfair solutions, especially if the outliers belong to subgroups that we aim to serve better.

To mitigate some of the challenges of ERM, tilted losses are used for the generalization of this traditional technique. Motivated in large by exponential tilting in deviation theory, works in Ref (Li et al., 2020; 2023) introduce TERM, an extension of ERM with an additional tilt hyperparameter $t$. The flexibility of tilted hyperparameter allows the model to continuously adjust decision boundaries based on the problem settings, offering robustness against outliers, fairness towards underrepresented subgroups, or a balance between both. This approach is especially useful in classification tasks where different groups or data distributions require varying levels of emphasis.

**Definition 1** (Empirical Risk Minimization — ERM (Vapnik, 1998)). *For a hypothesis $h(\mathbf{x}^{(i)})$, empirical risk minimization, the average loss over the training data, is defined as*

$$\bar{R}(\theta) := \frac{1}{N} \sum_{i \in [N]} \mathcal{L}(h(\mathbf{x}^{(i)}), y^{(i)}, \theta), \tag{1}$$

*where $\mathcal{L}(h(x_i), y_i, \theta)$ is the loss function that quantifies the distance between the prediction $h(x_i)$ and true label $y_i$ and parameter $\theta$, N is the number of training data points.*

While ERM focuses on minimizing the average loss over the training dataset, TERM introduces a modified approach by applying the exponential tilting technique to ERM, assigning different level of emphasis to the loss of samples.

**Definition 2** (Tilted Empirical Risk Minimization — TERM (Li et al., 2023)). *For $t \in \mathbb{R}^{\setminus 0}$, the t-tilted loss in ERM is defined as the tilted empirical risk minimization, given by*

$$\tilde{R}(t; \theta) := \frac{1}{t} \log \left( \frac{1}{N} \sum_{i \in [N]} e^{t\mathcal{L}(h(\mathbf{x}^{(i)}), y_i, \theta)} \right), \qquad (2)$$

*where $\mathcal{L}(h(\mathbf{x}^{(i)}), y_i, \theta)$ is the loss function on hypothesis $h(\mathbf{x}^{(i)})$, true label $y_i$ and parameter $\theta$, and N is the number of training samples.*

This framework introduces a flexible tilting tool to address the shortcomings of ERM, offering more robust solutions by adjusting the sensitivity of model to outliers using the tilt parameter.

## 2.2 QUANTUM NEURAL NETWORKS AND TRAINING

QNNs are a key class of QML models, which essentially combine concepts from classical neural networks and parameterized quantum circuits into hybrid architectures. Demonstrations of advantages of QNNs over modern deep-learning architectures remain open, given also the current hardware limitations . QNNs fundamentally provide input-output relationships via an exponentially-sized Hilbert space, which may allow advantages over classical counterparts in handling complex data and complex input-output relationships for specific scenarios. QNNs share a conceptually similar structure to classical neural networks, where a task is encoded into a parameterized loss function that is evaluated using a quantum computer, and a classical optimizer trains the parameters in the parameterized circuit.

A QNN is a function $f : \mathbb{C}^n \to \mathbb{C}^m$ that maps classical input data $\mathbf{x} \in \mathbb{C}^n$ to output data $\mathbf{y} \in \mathbb{C}^m$ through a parameterized quantum circuit $U(\boldsymbol{\theta})$, where $\boldsymbol{\theta} \in \mathbb{R}^p$ represents the parameters of the circuit. The process can be described as follows.

- **State preparation:** Consists of a feature map that encodes input data $\mathbf{x}$ into quantum states $|\psi_{\mathbf{x}}\rangle$. Here we assume the initial state to be the computational zero state $|\bar{0}\rangle$.

- **Circuit with a sequence of quantum gates:** This sequence of gates depends on input $\mathbf{x}$ and trainable parameters $\boldsymbol{\theta}$ (weights) optimized during the training process. The final state is obtained by applying a total unitary $U(\mathbf{x}; \boldsymbol{\theta})$ of the circuit onto the input state, i.e., $|\psi(\boldsymbol{\theta})\rangle = U(\mathbf{x}; \boldsymbol{\theta})|\bar{0}\rangle$.

- **Measurement:** This step determines the output of the quantum circuit by evaluating the model's performance. The prediction $y$ is obtained through $m$ measurement operators $M_i, i \in [m]$ on the corresponding qubits to derive expectation values $y_i = \langle\psi(\boldsymbol{\theta})|M_i|\psi(\boldsymbol{\theta})\rangle$.

- **Optimizer:** The measured values are used as a feedback to learn the best parameters $\boldsymbol{\theta}$. The optimization typically involves minimizing a loss function, which can be formulated as different forms of risk, such as ERM or TERM, as used in this work.

QNNs are you often explored in conjunction with implementations in near-term quantum computing architectures (Preskill, 2018).

Efficient QNN training remains an open question. To evaluate the gradients of PQCs on quantum hardware, several methods have been proposed. The parameter-shift rule is one of these methods, as introduced by (Mitarai et al., 2018; Schuld et al., 2019). This method efficiently computes the gradient by evaluating the circuit at two shifted parameter values, enabling the optimization of quantum models with low computational overhead.

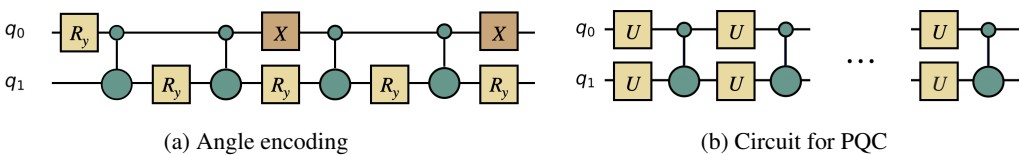

(a) Angle encoding          (b) Circuit for PQC

Figure 2: *Illustration of the circuits for angle encoding and the circuit.* (a) Here $R_y$ is the rotation operator that modifies the rotation axis $Y$ of the qubit by an angle $\theta$. $X$ is the Hadamard gate, and blue circles represent CNOT, where the smaller circle is the control qubit and the larger circle is the target qubit. Angle encoding method for data embedding into quantum states. Five preprocessed feature angles $\theta$ are used to encode each data point. (b) Here $U$ is the parameterized unitary used to encode the weights, where each $U$ takes 3 parameters as input.

## 3 NUMERICAL EXPERIMENTS

### 3.1 DATA AND ENCODINGS

**Datasets.** We focus on binary classification tasks. For this purpose, we adopt two datasets: the Iris flower dataset (Anderson, 1936; Fisher, 1936), as well as a synthetic dataset from Ref. (Li et al., 2020). After loading the dataset, data padding was adopted to make the data dimension same as the size of the quantum state vector. The final step of data pre-processing is the normalization, in order to encode the data into rotational angles of the rotational unitary gates. In this case, we need to encode the classical data of two real value points into the QNN for each use. Characteristics of each dataset are described below.

- **Iris dataset.** From the Iris, the two classes Setosa and Virsicolour were chosen. For the 2 dimensions, we used a scaled sepal width against sepal length. In the initial dataset, there are 50 data points in each class. To better show the effect of TERM, we want to create class imbalance and an outlier. Class Setosa was selected as the majority class and kept as is. To create imbalanced classes, we specifically select 5 data points randomly from the Virsicolour dataset, to make it the minority class. Furthermore, one of the 5 data points was selected as the outlier and edited to behave like a Setosa data point. Originally, four features were measured from each sample: the length and the width of the sepals and petals. We selected only 2 features — the length and the width of the sepals.

- **Synthetic dataset.** This dataset is manually randomly generated. There are 56 data points in the majority class, and 6 data points in the minority class (including 1 outlier), adding up to a total of 62 data points.

**Angle Encoding.** To train a QNN using classical data, we must first convert that data into a quantum representation. This conversion is accomplished through an encoding process, which we include in our numerical results. The encoding transforms classical data points into quantum states. Here we discuss angle encoding which is a widely used technique in representing classical data in on a quantum computer in the context of machine learning (LaRose & Coyle, 2020; Schuld & Petruccione, 2021). Angle encoding uses a sequence of controlled NOT (CNOT) gates and uniformly controlled rotations (Mottonen et al., 2004).

Specifically, in angle encoding, classical data is encoded into quantum gates using a rotation operator such as $R_x(\theta)$, $R_y(\theta)$ or $R_z(\theta)$, where $\theta$ represents the feature value of input data. For example, a 2-dimensional data point $x = [x_1, x_2]$ can be mapped into a quantum state by applying rotations on qubits, where $\theta = f(x)$ is some function of $x$ (e.g., direct or normalized values of $x_1, x_2$). Additionally, CNOT gates are used alongside with rotation gates to create entanglement between qubits. By applying these gates, the qubits are prepared in a state where their amplitudes encode the classical data features. These quantum states representing the encoded classical data, can then be processed by the variational quantum circuit for tasks such as classification or optimization. The circuit used to encode the data is shown in Fig. 2(a).

| Tilt hyperparameter | 20 | 10 | 5 | 1 | 0 (ERM) | -5 |
|---|---|---|---|---|---|---|
| **Iris** | **98.2%** | **98.2%** | 96.4% | 92.7% | 90.9% | 90.9% |
| **Synthetic** | **98.4%** | **98.4%** | **98.4%** | **98.4%** | 90.3% | 90.3% |

Table 1: *Accuracy of the QNN classification tasks on two datasets.*

## 3.2 QUANTUM MODEL

To demonstrate the effect of the tilted hyperparameter in QML models, we adopted a simple unitary model circuit with relatively few trainable parameters, inspired by Ref. (Schuld et al., 2020). This 2-qubit model is composed of three parts. Firstly, the data is encoded into quantum state using the angle encoding technique as shown above. Secondly, trainable parameters are embedded into the circuit. The circuit contains single-qubit rotation gates and fixed two-qubit CNOT gates. A layer in this quantum circuit consists of a parameterized unitary on each qubit followed by a entangling CNOT gate. In our model, 6 of these layers were used for the circuit. Randomly generated parameters will be substituted into the rotation gates initially, and will be updated by the classical optimizer to minimize the loss function. The number of trainable parameters is $6 \times 2 \times 3$ (*number of layers $\times$ number of qubits $\times$ number of parameters in each unitary*). The circuit we used is shown in Fig. 2(b). Lastly, in the read-out module, the expectation value of the computational basis measurement of the first qubit is used to determine the prediction of the binary classification problem. For the classical gradient update portion, we used gradient-descent optimizer with Nesterov momentum, an optimization method that improves the convergence rate for convex optimization problems from the traditional gradient descent's rate of $O(1/k)$ to $O(1/k^2)$ (Nesterov, 1983). Batch gradient descent was used with a batch size of 5, and the optimization process is halted when no improvement in accuracy is seen over 10 iterations. Square loss is used with the TERM function in Definition 2.

## 3.3 NUMERICAL RESULTS

Here we detail the results of our numerical experiments and denote the influence of a tilted hyperparameter in training a QNN.

For the first numerical experiments, we demonstrate the effect of the tilt hyperparameter on the decision boundaries of QNN classification tasks with imbalanced classes. We use a simple dataset with only two dimensions, namely the Iris dataset (Anderson, 1936; Fisher, 1936).

The final outcome of the first numerical experiment is shown in Fig. 3(a). It can be clearly seen that in the presence of class imbalance and outliers, the QNN classification task on the Iris dataset can be successfully tuned with the tilt hyperparameter. When $t$ is greater than 0, it can mitigate the effect of class imbalance and outliers, resulting in better decision boundaries compared to ERM ($t = 0$) and $t < 0$ cases.

In the second numerical experiment, illustrated in Fig. 3(b), we further aim demonstrate the ability of the tilt hyperparameter to boost the performance of QNN classification tasks. Here, we use the synthetic dataset from the original TERM result in Ref. (Li et al., 2020). Considering that this dataset is more complex than the Iris dataset, we expected the tilt to behave differently. From the results, we can see that for this case, non-tilted case give the decision boundary which misclassified several points in majority class. With tilt $t > 0$ in the loss function, this model pushes the boundary in correct direction and hence boosts the performance.

As shown in Table 1, with the fine-tuned tilt hyperparameter, we achieve better performance in QNN classification tasks compared with the non-tilt one ($t = 0$). Notably, in these two QNN classification tasks, with large positive tilt, we are able to achieve the best possible performance. As shown in Fig. 3, outliers cannot be accurately classified into their correct corresponding classes, however, the tilt hyperparameter remedies the outlier presence by reducing their importance in the decision boundary.

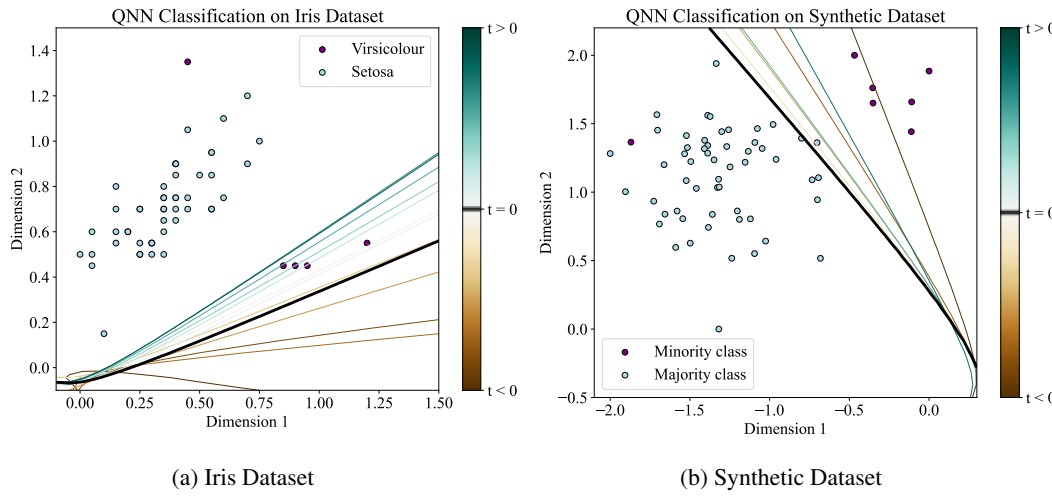

(a) Iris Dataset  (b) Synthetic Dataset

Figure 3: *The effects of TERM on the decision boundary as a function of $t$.* Binary classification for a 2-class 2-feature task on two different datasets. When $t = 0$, the original ERM objective is recovered, as highlighted in black.

## 4 TRAINING QNNS WITH TILTED QUANTUM GRADIENTS

While the training of QNNs discussed in Section 3 uses the standard classical optimization technique of gradient descent, this section explores a more advanced approach by applying quantum gradients to develop a hybrid quantum-classical optimization algorithm, a promising direction in near-term quantum algorithms (Preskill, 2018; Bharti et al., 2022). Here, we define and formalize the algorithm for a tilted parameter-shift rule.

### 4.1 TILTED PARAMETER-SHIFT RULE

In the optimization process of QNNs, we vary parameters to minimize the objective function. One approach to achieve this minimization is through the use of quantum gradients, with the parameter-shift rule being a commonly used technique for computing these gradients (Mitarai et al., 2018). This rule has been extended to more general cases, such as its application to quantum gates with more than two distinct eigenvalues and continuous-variable quantum circuits (Schuld et al., 2019), to a broader class of quantum gates, including multi-parameter and higher-order gates (Wierichs et al., 2022) and to arbitrary gates by decomposing gates into a product of standard gates Crooks (2019) and to any multi-qubit quantum evolution Banchi & Crooks (2021).

The working principle of parameter-shift rule is as follows. The unitary transformation performed by the variational circuit can be decomposed into a product of unitary operations,

$$U(\mathbf{x}; \boldsymbol{\theta}) = U_N(\theta_N)U_{N-1}(\theta_{N-1})\cdots U_i(\theta_i)\cdots U_1(\theta_1)U_0(x) = \prod_{j=N}^{1} U_j(\theta_j)U_0(\mathbf{x}), \quad (3)$$

where each gate takes the form as $U_j(\gamma_j) = e^{i\gamma_j H_j}$, and $H_j$ is a Hermitian operator. The expectation value of measuring an observable $O$ on the output state of this computation can be seen as an objective function $f$, that depends on the input $\mathbf{x}$ and the parameter vector $\boldsymbol{\theta}$, and is given by

$$f(\mathbf{x}; \boldsymbol{\theta}) = \langle 0|U^\dagger(\mathbf{x}; \boldsymbol{\theta})OU(\mathbf{x}; \boldsymbol{\theta})|0\rangle = \langle \mathbf{x}| \prod_{j=1}^{N} U_j^\dagger(\theta_j)O \prod_{j=N}^{1} U_j(\theta_j)|\mathbf{x}\rangle, \quad (4)$$

where $|\mathbf{x}\rangle = U(\mathbf{x})|0\rangle$. It brings us to the definition of the gradient of the objective function, which is the so-called parameter-shift rule. See (Mitarai et al., 2018; Schuld et al., 2019) for more technical details.

**Definition 3** (Parameter-shift rule). *The parameter-shift rule states that the derivative of the objective function of a quantum circuit with respect to a gate parameter $\boldsymbol{\theta}$ is*

$$\nabla_{\boldsymbol{\theta}} f(\mathbf{x}; \boldsymbol{\theta}) = c[f(\mathbf{x}; \boldsymbol{\theta} + s) - f(\mathbf{x}; \boldsymbol{\theta} - s)], \tag{5}$$

*where $c$ is a shift constant and $s$ is the parameter-shift both depending on the eigenvalues of operators $H$.*

This approach computes gradients efficiently by requiring only two circuit evaluations for each parameter.

However, in the presence of imbalanced datasets, using weighted losses allows for assigning higher weights to the minority class, which improves the models' ability to distinguish between the two different classes. TERM assigns different weights to data points based on their loss values, allowing the model to adjust its parameters based on the collective information from the dataset.

For the gradients of TERM, there is an essential assumption that the loss function is continuously differentiable with respect to parameters, as Assumption 1 in Ref. (Li et al., 2023). This assumption ensures the existence of the gradient of the loss function and allows applying this loss function to the tilted gradient described in the following theorem.

**Theorem 4.1** (Tilted gradient (Li et al., 2023)). *Given a smooth loss function $l(x; \theta)$, the gradient of TERM with respect to $\theta$ is*

$$\nabla_{\theta} \tilde{R}(t; \theta) = \sum_{i \in [N]} \tilde{w}_i(t; \theta) \nabla_{\theta} l(\mathbf{x}^{(i)}; \theta), \tag{6}$$

*where $\tilde{w}_i$ is the tilted weight given by*

$$\tilde{w}_i(t; \theta) := \frac{e^{tl(\mathbf{x}^{(i)}; \theta)}}{\sum_{j \in [N]} e^{tl(\mathbf{x}^{(j)}; \theta)}} = \frac{1}{N} e^{t(l(\mathbf{x}^{(i)}; \theta) - \tilde{R}(t; \theta))}, \tag{7}$$

*$l(\mathbf{x}^{(i)}; \theta)$ is the loss function of input $\mathbf{x}^{(i)}$ and parameters $\theta$, and $t$ is tilt hyperparameter.*

The theorem demonstrates that the tilted gradient is a weighted average of the individual losses gradients, with each data point assigned a weight grows exponentially with its corresponding loss value.

Based on this theorem, applying tilting techniques to the parameter-shift rule extends the scenario of quantum circuits to handle multiple weighted inputs cases, allowing for more efficient gradient computation in complex quantum machine learning tasks, specifically for classification.

**Definition 4** (Tilted parameter-shift rule). *The tilted batch gradient of an objective function defined by a parameterized quantum circuit is given by:*

$$\nabla_{\theta} \mathcal{G}(t; \boldsymbol{\theta}) := \sum_{i \in [N]} w_i(t; \boldsymbol{\theta}) \nabla_{\boldsymbol{\theta}} g(\mathbf{x}^{(i)}; \boldsymbol{\theta}) = c \sum_{i \in [N]} w_i(t; \boldsymbol{\theta}) [g(\mathbf{x}^{(i)}; \boldsymbol{\theta} + s) - g(\mathbf{x}^{(i)}; \boldsymbol{\theta} - s)], \tag{8}$$

*where the tilted weight is defined as:*

$$w_i(t; \boldsymbol{\theta}) := \frac{e^{tg(\mathbf{x}^{(i)}; \boldsymbol{\theta})}}{\sum_{j \in [N]} e^{tg(\mathbf{x}^{(j)}; \boldsymbol{\theta})}}, \tag{9}$$

*with $\mathbf{x}^{(i)}$ representing the input data points, $g(\mathbf{x}^{(i)}; \boldsymbol{\theta})$ being the smooth objective function evaluated at $\mathbf{x}^{(i)}$ and parameter $\boldsymbol{\theta}$, $c$ as a shift constant and $s$ as the parameter shift corresponding to $g(\mathbf{x}^{(i)}; \boldsymbol{\theta})$.*

**Remark 1.** *Since the tilted weight is a normalization factor, the tilted gradient can be viewed as an average of the gradient of $g(\mathbf{x}; \boldsymbol{\theta})$ among all data points.*

By combining the parameter-shift rule with the TERM framework, we can optimize the parameters of quantum circuits with different weighted inputs effectively. The tilted parameter-shift rule allows us to better handle of outliers or difficult data points, assigning more focus to specific inputs during optimization. Many machine learning tasks, particularly classification (Farhi & Neven, 2018), rely on optimizing performance by balancing the weighted contributions of various inputs. When each input is weighted differently based on its importance or reliability, the model must effectively balance these weights to achieve accurate predictions. This becomes especially important in tasks involving imbalanced datasets, noisy data, or when certain input features carry more significance.

## 4.2 IMPLEMENTATION OF QNN TRAINING

---

**Algorithm 1** QNNs Mini-Batch Training with Tilted Quantum Gradient

---

**Require:** Quantum circuit $U(\mathbf{x}; \boldsymbol{\theta})$ with input $\mathbf{x}$ and parameters $\boldsymbol{\theta} = (\theta_1, \theta_2, \ldots, \theta_n)$, training dataset $\mathcal{D}$, measurement observable $O$, learning rate $\eta$, Parameter-shift $s$, shift constant $c$, tilt hyperparameter $t$ and limit of iterations $R$

1: Initialize parameters $\boldsymbol{\theta}$ randomly and the iteration $K \leftarrow 0$
2: **if** $K < R$ **then**
3:   Initialize the tilted weight sum $W(\boldsymbol{\theta}) = 0$ and the weighted gradient $\nabla_{\boldsymbol{\theta}} \mathcal{G} = 0$
4:   Sample $d$ data points from a mini batch $\mathcal{B} \subset \mathcal{D}$ randomly
5:   **for** $i = 1$ to $d$ **do**
6:     Compute the objective functions $g(\mathbf{x}^{(i)}; \boldsymbol{\theta})$
7:     Compute the exponential loss $w_i(\boldsymbol{\theta}) = e^{tg(\mathbf{x}^{(i)}; \boldsymbol{\theta})}$
8:     Update the tilted weight sum $W(\boldsymbol{\theta}) \leftarrow W(\boldsymbol{\theta}) + w_i(\boldsymbol{\theta})$
9:     Compute the value of $\nabla_g \mathcal{L}(\boldsymbol{\theta})$
10:   **end for**
11:   **for** $i = 1$ to $d$ **do**
12:     Normalize the tilted weight $w_i(\boldsymbol{\theta}) \leftarrow \frac{w_i(\boldsymbol{\theta})}{W(\boldsymbol{\theta})}$
13:     **for** $j = 1$ to $n$ **do**
14:       Evaluate the objective function with shifted parameters:

$$g_+(\mathbf{x}^{(i)}; \theta_j) = g\left(U(\mathbf{x}^{(i)}; \theta_1, \ldots, \theta_j + s, \ldots, \theta_n)\right)$$

$$g_-(\mathbf{x}^{(i)}; \theta_j) = g\left(U(\mathbf{x}^{(i)}; \theta_1, \ldots, \theta_j - s, \ldots, \theta_n)\right)$$

Compute individual quantum gradient estimate via parameter-shift rule:

$$\nabla_{\theta_j} g(\mathbf{x}^{(i)}; \theta_j) \leftarrow c\left(g_+(\mathbf{x}^{(i)}; \theta_j) - g_-(\mathbf{x}^{(i)}; \theta_j)\right)$$

15:     **end for**
16:     Compute the weighted quantum gradient for each input via tilted parameter-shift rule:

$$\nabla_{\boldsymbol{\theta}} \mathcal{G}(\boldsymbol{\theta}) \leftarrow \nabla_{\boldsymbol{\theta}} \mathcal{G}(\boldsymbol{\theta}) + w_i(\boldsymbol{\theta}) \nabla_{\boldsymbol{\theta}} g(\mathbf{x}^{(i)}; \boldsymbol{\theta})$$

17:     Evaluate the gradient $\nabla_{\boldsymbol{\theta}} \mathcal{L}(\boldsymbol{\theta}) \leftarrow \nabla_g \mathcal{L}(\boldsymbol{\theta}) \nabla_{\boldsymbol{\theta}} \mathcal{G}(\boldsymbol{\theta})$
18:     Update parameters using gradient descent: $\boldsymbol{\theta} \leftarrow \boldsymbol{\theta} - \eta \nabla_{\boldsymbol{\theta}} \mathcal{L}(\boldsymbol{\theta})$
19:   **end for**
20:   Update the iteration $K \leftarrow K + 1$
21: **end if**
**Ensure:** Optimized parameters $\boldsymbol{\theta}^\star$

---

In this case, we aim to classify data using a QNN where the weights assigned to different inputs influence how the model processes and learns from data. These weights affect how much each input contributes to the overall prediction, and the learning algorithm must adjust the model parameters accordingly to minimize the classification error while taking these weights into account.

Firstly, let the least-squares objective be the loss function evaluated at parameter $\boldsymbol{\theta}$ and input $(\mathbf{x}^{(i)}, y_i)$ given by the training set $\mathcal{D} = \{(\mathbf{x}^{(1)}, y_1), \cdots, (\mathbf{x}^{(N)}, y_N)\}$,

$$\mathcal{L}(\boldsymbol{\theta}) = \sum_i^N \left| g(\mathbf{x}^{(i)}; \boldsymbol{\theta}) - y_i \right|^2,$$

when the objective function $g(\mathbf{x}^{(i)}; \boldsymbol{\theta})$ is predicted by the variational circuit. To optimize the parameters $\boldsymbol{\theta}$, we compute the gradient of the loss function with respect to $\boldsymbol{\theta}$. Applying the matrix product of the general chain rule, the gradient is expressed as

$$\nabla_{\boldsymbol{\theta}} \mathcal{L}(\boldsymbol{\theta}) = \frac{\partial \mathcal{L}(\boldsymbol{\theta})}{\partial g(\mathbf{x}^{(i)}; \boldsymbol{\theta})} \frac{\partial g(\mathbf{x}^{(i)}; \boldsymbol{\theta})}{\partial \boldsymbol{\theta}},$$

where $\frac{\partial \mathcal{L}(\boldsymbol{\theta})}{\partial g(\mathbf{x}^{(i)};\boldsymbol{\theta})}$ can be easily calculated by the classical optimizer and quantum gradient $\frac{\partial g(\mathbf{x}^{(i)};\boldsymbol{\theta})}{\partial \boldsymbol{\theta}}$ can be computed using methods, like parameter-shift rule.

For large datasets, evaluating the circuit twice for each parameter and each data point may become computationally expensive. To address it, we employ mini-batch method for training. At every iteration, we sample a batch of data points, $\mathcal{B} \subset \mathcal{D}$, and run the circuit without shift to obtain the objective function $g(\mathbf{x}^{(i)};\boldsymbol{\theta}) = \langle 0|U^{\dagger}(\mathbf{x}^{(i)};\boldsymbol{\theta})OU(\mathbf{x}^{(i)};\boldsymbol{\theta})|0\rangle$. Next we focus on evaluating the quantum gradient. To balance the effect of outliers and smoother gradient estimation, we apply tilted parameter-shift rule, defined in Definition 4, with $\frac{\partial g(\mathbf{x}^{(i)};\boldsymbol{\theta})}{\partial \boldsymbol{\theta}}$ replaced by the tilted average $\nabla_{\theta}\mathcal{G}(t;\boldsymbol{\theta})$. Finally, the parameters are updated using gradient descent with learning rate $\eta$. The iteration ends until convergence or a predefined stopping criterion is met, like reaching the limit of number of iterations $R$. We present our algorithm for QNNs training in Algorithm 1.

## 5 DISCUSSION AND CONCLUSION

This work marks a first step in understanding the feasibility of the tilted term in training quantum learning models. Here, we have demonstrated the analytical performance of a loss function containing the tilted parameter, under the parameter-shift rule. More specifically, we define gradient training for parameter-shift rule using the tilted parameter. On the numerics front, we have presented the performance of the loss function with tilt on two different datasets, namely the Iris and a synthetics dataset from the original TERM work (Li et al., 2023). Our results have evidenced the role of TERM in training a class of QNNs for two class imbalance induced datasets. For the Iris, positive tilted hyperparameters mitigate the effect of imbalanced data class. While for the synthetic dataset the non-tilted case, which correspond to ERM, has the worst performance compared to tilted cases, thus both positive and negative tilted parameters boost the performance of QNN classification tasks.

In future work, we propose benchmarking our numerical analysis further against more complex datasets and additional sophisticated models. From a theoretical perspective, it would also be interesting to see the effect of titled loss in quantum models with respect to the generalization property of these models (Caro et al., 2022). Heuristically, the addition of regularizers terms to the training process should, in principle, aid generalization. Moreover, tilted parameter-shift rule inspires us to define the quantum version of TERM (QTERM) by introducing objective function of quantum circuits. The necessity for QTERM arises from quantum alternative of the loss function. This could lead to interesting future research directions on quantum risk akin to the ERM principle.

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
