# OpenReview forum: "Tilted Losses in Training Quantum Neural Networks"
_ICLR.cc/2025/Conference — ICLR 2025 Conference Withdrawn Submission_

### Official Review · Reviewer_W5Xp · 2024-10-30

**Soundness:** 2
**Presentation:** 2
**Contribution:** 1
**Rating:** 3
**Confidence:** 3

**Summary:**

This paper applies a tilted loss to the training of QNN on binary classification. The paper also proposes a tilted parameter-shift rule to update the parameters when using tilted loss.

**Strengths:**

There's limited research in QML compared to classical ML algorithms. Improving QML methods is often nontrivial given the computation constraints, and attempts to adapt methods from classical ML to QML are challenging but important.

**Weaknesses:**

1. Novelty/originality: I have a very hard time seeing the contribution of this paper. The conclusions of applying tilted loss were very similar to the original paper, and combining tilted loss and the parameter-shift rule does not seem novel.

2. Significance: If the contribution of the paper is empirical instead of methodological, the paper also only ran experiments on toy datasets. The discussion about decision boundary is also not convincing given that there are only two input dimensions. What does it even mean to have outliers for binary classification, in this case? That is, from these two features, there's no reason to believe an outlier of class A is not class B. Such discussion seems to only make sense when the input is complicated enough.

3. The writing does not convey a clear message. It also uses "Ref." to refer to a lot of different refernces, which is kind of weird. Multiple typos here and there as well (e.g. L210 "are you"). There are also places like L255-260 where an entire paragraph reads like the rephrasing of the same sentence in many repetitions.

**Questions:**

1. What's the main challenge in applying tilted loss in QNN?

2. Did the authors run any experiments on the proposed tilted parameter-shift rule?

---

### Official Review · Reviewer_aGNH · 2024-10-31

**Soundness:** 1
**Presentation:** 2
**Contribution:** 1
**Rating:** 3
**Confidence:** 4

**Summary:**

The authors propose employing a tilted loss function in training Quantum Neural Networks, where each data point is reweighted with a softmax-like term (with temperature parameter t, and logit given by the standard loss term associated with the data point). The aim is to help the model better adjust to scenarios where data imbalances and outliers are present in the dataset. The authors claim an improvement of the decision boundaries and training accuracies in the empirical tests presented in the paper.

**Strengths:**

Moving beyond the i.i.d hypothesis in the loss functions of neural networks is an interesting research direction. Discussing the impact of heuristics like the Tilted ERM in the training of quantum neural networks could be useful if these devices are to be used in practice in the future.

**Weaknesses:**

The paper mostly discusses and presents methods and ideas introduced in previous literature. The novel part, on exploring the new TERM method in the training of QNN, is too shallow to allow any assessment of the validity of the method. The only empirical results are based on two simulations on hand-crafted instances of small datasets (the second is not detailed in the present paper). No statistics are provided, which should prevent any claim of robustness of the method. The TERM method itself seems to be impractical, due to the necessity of computing the normalization of the gradient reweighting coefficients, at each parameter update (and is therefore incompatible with mini-batch training). The authors only discuss the impact on training accuracy on the two analyzed tasks, focusing on optimization and leaving the impact on generalization for future work. Moreover, the interpretation of the authors that the tilted loss would reduce the importance of outliers, or give different weights to more or less reliable data, seems to be at odds with the effect of the softmax reweighting term, which gives larger weight to the data points where the network is currently making the larger errors. The claim that both positive and negative tilt temperatures lead to an improvement seems to be at odds with the results in Table 1.

**Questions:**

1) The authors should drastically expand their empirical results section. Running simulations on two small hand-crafted datasets cannot be presented as sufficient evidence of the effectiveness of an optimization method. Moreover, it is not clear why the standard term leads the model to misclassify points that are basically linearly separable.
2) Discussing the impact on generalization of the prescribed optimization method would possibly be the most interesting part of this work. Given that most of the paper is devoted to repetitions and to discussion of non-original content, it would seem reasonable to ask the authors to make space for these explorations within this work.
3) The reweightings of the TERM objective are not fixed once and for all, but evolve during the optimization. The authors should adjust accordingly their claims on their method reducing the relevance of outliers and more/less represented points in the dataset (since after a while, when the rest of the data points are already well-fitted, the weight will concentrate on the points where the loss is still high, e.g. the misclassified outlier).
4) Can the authors discuss a heuristic for fixing the tilt parameter? It would appear that fitting it on the training accuracy might lead to overfitting problems.
5) Overall, the redundancy of the presentation needs to be reduced in favor of further experiments.

---

### Official Review · Reviewer_GKMP · 2024-11-03

**Soundness:** 2
**Presentation:** 3
**Contribution:** 2
**Rating:** 5
**Confidence:** 4

**Summary:**

The paper investigates the integration of tilted empirical risk minimization (TERM) into the training of quantum neural networks (QNNs) to enhance their robustness against outliers and improve training accuracy. By introducing a tilted hyperparameter into the loss function, the authors aim to penalize outliers more effectively during the training process of QNNs, specifically for binary classification tasks. They extend the parameter-shift rule—a common method for training QNNs—to accommodate this tilted loss function and demonstrate its impact through experiments on both the Iris dataset and a synthetic dataset with induced class imbalance.

**Strengths:**

he paper aims to extend the concept of tilted empirical risk minimization (TERM) from classical machine learning (ML) to quantum machine learning (QML), specifically for training quantum neural networks (QNNs). It investigates the effects of a tilted loss function in QNN training for binary classification tasks, particularly in the presence of class imbalance.

**Weaknesses:**

The authors primarily adapt an established technique—tilted empirical risk minimization (TERM)—to the framework of quantum neural networks (QNNs) without introducing significant modifications or new insights. This approach raises concerns regarding the originality of the work. While extending existing methodologies to new contexts can be valuable, the paper does not sufficiently demonstrate how the adaptation contributes to the advancement of knowledge in quantum machine learning. The lack of innovative modifications or theoretical developments may lead to a perception of limited originality in the contributions presented.

**Questions:**

1. The evaluation is primarily based on the Iris dataset and a synthetic dataset. Have you considered testing your approach on a broader range of datasets with varying complexities and characteristics? This could help validate the robustness of your findings.

2 In Function 9, there is a concern regarding the possibility of the denominator becoming zero, which would result in divergence and prevent the function from converging. It is crucial to address this issue, as it could lead to undefined behavior during the optimization process.

3 Given the hyperparameters in your proposed method, why did you choose not to conduct a parameter search to optimize these settings? How do you ensure that the selected hyperparameters are the best for the performance of your algorithm?

4 The testing results presented appear to be limited. Can you provide additional experimental results or evaluations to better support the effectiveness of your algorithm?

5. From the results, it appears that higher values of the parameter t lead to increased accuracy. However, I am concerned that as t increases, both the computational expense and the risk of encountering out-of-range values also increase. How do you define the optimal limit for t? What criteria or methods do you use to determine a suitable range for t that balances accuracy with computational feasibility and stability?

---

### Note · Authors · 2024-11-13

I have read and agree with the venue's withdrawal policy on behalf of myself and my co-authors.